# Back to Nature: Medicinal Plants as Promising Sources for Antibacterial Drugs in the Post-Antibiotic Era

**DOI:** 10.3390/plants12173077

**Published:** 2023-08-28

**Authors:** Emad M. Abdallah, Bader Y. Alhatlani, Ralciane de Paula Menezes, Carlos Henrique Gomes Martins

**Affiliations:** 1Department of Science Laboratories, College of Science and Arts, Qassim University, Ar Rass 51921, Saudi Arabia; 140208@qu.edu.sa; 2Unit of Scientific Research, Applied College, Qassim University, Buraydah 52571, Saudi Arabia; 3Technical School of Health, Federal University of Uberlândia, Uberlândia 38400-732, MG, Brazil; ralciane@ufu.br; 4Laboratory of Antimicrobial Testing, Federal University of Uberlândia, Uberlândia 38405-320, MG, Brazil; carlos.martins2@ufu.br

**Keywords:** antibacterial activity, antibiotic-resistant, phytochemicals, synthetic, mechanism of action, infections

## Abstract

Undoubtedly, the advent of antibiotics in the 19th century had a substantial impact, increasing human life expectancy. However, a multitude of scientific investigations now indicate that we are currently experiencing a phase known as the post-antibiotic era. There is a genuine concern that we might regress to a time before antibiotics and confront widespread outbreaks of severe epidemic diseases, particularly those caused by bacterial infections. These investigations have demonstrated that epidemics thrive under environmental stressors such as climate change, the depletion of natural resources, and detrimental human activities such as wars, conflicts, antibiotic overuse, and pollution. Moreover, bacteria possess a remarkable ability to adapt and mutate. Unfortunately, the current development of antibiotics is insufficient, and the future appears grim unless we abandon our current approach of generating synthetic antibiotics that rapidly lose their effectiveness against multidrug-resistant bacteria. Despite their vital role in modern medicine, medicinal plants have served as the primary source of curative drugs since ancient times. Numerous scientific reports published over the past three decades suggest that medicinal plants could serve as a promising alternative to ineffective antibiotics in combating infectious diseases. Over the past few years, phenolic compounds, alkaloids, saponins, and terpenoids have exhibited noteworthy antibacterial potential, primarily through membrane-disruption mechanisms, protein binding, interference with intermediary metabolism, anti-quorum sensing, and anti-biofilm activity. However, to optimize their utilization as effective antibacterial drugs, further advancements in omics technologies and network pharmacology will be required in order to identify optimal combinations among these compounds or in conjunction with antibiotics.

## 1. Introduction

Since time immemorial, humans (*Homo sapiens*) have recognized that some plants possess curative properties for specific ailments. Fossil records indicate that a human being used a therapeutic herb called hollyhock (*Alcea rosea* L.) in Mesopotamia (Iraq) 60,000 years ago [1,2]. However, man may have used medicinal plants for a long time before that, as the modern human lineage originated in Africa 300,000 years ago [3,4]. During that period, there was a lack of comprehensive knowledge regarding the causes of illnesses and the specific plants that could be utilized as remedies. Consequently, the utilization of medicinal plants relied heavily on empirical evidence. Over time, the underlying reasons behind the effectiveness of certain medicinal plants in treating specific diseases started to be unraveled, leading to a shift from an empirical framework to a more evidence-based approach [5]. Until the emergence of iatrochemistry in the 16th century, plants served as the primary source for both the treatment and prevention of ailments. However, with the diminishing efficacy of synthetic drugs and the increasing number of contraindications associated with their usage, the relevance of natural remedies is once again in the spotlight. The renewed focus on natural drugs has been driven by the need for alternative treatments that can address contemporary challenges [6,7]. Regarding antibacterial drugs, the discovery of penicillin by Fleming in 1928, coupled with the subsequent exploration and clinical application of sulfonamides in the 1930s, marked the advent of modern antibiotherapy [8]. Penicillin swiftly gained widespread usage during the early 1940s. By the 1950s, the era commonly referred to as the “golden era” of antibiotic development and utilization was well underway, with the introduction of multiple novel antibiotic classes until the 1970s [9]. Nevertheless, subsequent to that illustrious period, a gradual escalation in antibiotic resistance has transpired, eventually culminating in a formidable global crisis. This predicament has been primarily fueled by the overutilization and inappropriate employment of antibiotics, alongside a dearth of novel drug development by the pharmaceutical industry [10]. Recently, in 2023, the global prevalence of antibiotic-resistant bacteria has become a cause for widespread concern. This disconcerting trend is further compounded by the absence of new antibiotic classes being developed, ultimately giving rise to what is commonly referred to as the “antibacterial crisis” [11]. With the rise of this global problem, medicinal plants have garnered considerable attention as a promising source for the discovery of new antibacterial drugs. These plants possess a wide array of chemical constituents, such as alkaloids, flavonoids, terpenoids, and phenolic compounds, which exhibit diverse biological activities, including potent antibacterial properties [12]. The objective of this review was to explore the issue of antibiotic resistance from a historical perspective and compile current knowledge from scientific publications regarding the potential of medicinal plants as an alternative approach. The field of research was focused on novel antibacterial drug development using various medicinal plants. The review covers multiple aspects, including the antibiotic dilemma, the reasons behind antibiotic collapse, and the emergence of antibiotic-resistant bacteria. It also discusses the significance of medicinal plants as powerful alternatives to antibiotics, highlighting the mechanisms of action of the bioactive phytochemicals present in these plants, as well as the advantages of utilizing plant-based antibacterial molecules alone or in combination with antibiotics or different phytochemicals. The review acknowledges the challenges and obstacles faced in the discovery of plant-based drugs, and it provides future perspectives on this subject.

## 2. Methodology

To conduct the review, a search was conducted across several scientific databases, including PubMed, Elsevier, ResearchGate, Scopus, and Google Scholar. The search encompassed publications from 1934 to 2023. A total of 152 publications were collected and thoroughly assessed. Moreover, the number of published reports on the antibacterial properties of medicinal plants from 2012 to 2022 was counted from Google Scholar and graphed. Additionally, a survey was conducted specifically in the PubMed databases to identify publications evaluating the antibacterial action of plant extracts against WHO priority species. The search of these databases used English MeSH descriptors, and articles published between 2018 and 2022 were included. Duplicate articles, reviews, and those evaluating isolated, commercially acquired molecules were excluded to ensure the selection of relevant studies.

## 3. The Pre-Antibiotic Era

Pathogenic bacteria have had an impact on human demography and culture on Earth over the ages via illnesses, epidemics, and pandemics. Numerous epidemics and pandemics have been documented or are thought to have happened throughout human history, although the majority of their causative agents remain unknown, and we are sometimes unable to determine whether they are caused by a particular bacterium or virus. It is critical to emphasize here that there is some degree of collaboration between viral and bacterial pathogens in creating epidemics or even pandemics, and this topic is often neglected. For example, over 50 million people died during the 1918 influenza pandemic (Spanish flu), where certain fatalities looked to be caused by viral pneumonia, since they happened promptly after the beginning of symptoms, often accompanied by abrupt lung bleeding or edema. Clinical and pathological data, on the other hand, revealed that a majority of victims died of subsequent bacterial pneumonia [13,14].

Almost all major global civilizations throughout history, from ancient Egypt, Mesopotamia, Babylon, ancient Rome, ancient China, and ancient India to the great Islamic civilizations, have left written records of devastating large-scale epidemics that have occurred [15,16,17,18]. However, determining exactly what type of microorganism was responsible for a given epidemic in ancient times is problematic.

Interestingly, the early modern age saw many deadly epidemics and pandemics, the majority of which were bacterial in origin. Several of these terrible epidemics that were diagnosed as bacterial diseases are shown in Figure 1.

The black death (bubonic plague), which spread between 1346 and 1353, is a bacterial infection caused by Yersinia pestis and transmitted by fleas that feed on rats. It afflicted mankind and wiped out 40% of Europe’s population, with total estimates ranging from 50 million to 200 million fatalities worldwide [20,21]. Syphilis, commonly known as “the great pox”, is an infection caused by the bacterium Treponema pallidum. Symptoms include exploding boils and decaying flesh. The pandemic, which began during the French siege of Naples in 1494, was a horrific sickness that swept over Europe, killing approximately 5 million people [22,23]. In 1817, a new terrible epidemic broke out in India and soon spread across Asia, Africa, and Europe, lasting until 1823; there is no precise figure for the number of deaths caused by the cholera pandemic, although it is thought to be in the hundreds of thousands [24,25]. Diphtheria (caused by the bacterium *Corynebacterium diphtheriae*) had reached epidemic levels in 1880, and the death toll had risen substantially; the average case fatality ratio had risen to more than 50%, with the bulk of the victims being children [26,27]. By the 1900s, TB (*tuberculosis*), caused by *Mycobacterium tuberculosis*, had spread rapidly throughout Eastern Europe, Africa, Asia, and South America; this infection has always been liked to a high mortality rate, and Koch stated in 1901 to the British Tuberculosis Congress that “there is no disease which inflicts such deep wounds on mankind as this” [28,29,30]. Interestingly, the 1918 viral pandemic (Spanish flu), which killed over 50 million people, was discovered to be associated with a lethal bacterial pneumonia, as many contemporaneous scientists observed. This unusually fatal flu (Spanish flu) did not kill people in large numbers, but the virus enabled colonizing strains of bacteria to cause highly lethal pneumonia [19,31,32].

## 4. The Golden Era of Antibiotics

Historically, antibiotics have undeniably reduced the spread of bacterial epidemics. Throughout the golden period of antibiotics (from 1930 to 1960) and the subsequent innovation gap (between the 1960s and the 2000s), pathogenic bacteria had no apparent influence on human history, as no severe bacterial pandemic was documented, with the exception of multiple global outbreaks of antibiotic-resistant bacteria, e.g., the Aberdeen typhoid outbreak of 1964 in Britain [33], an outbreak of Sverdlovsk anthrax in the Soviet Union in 1979 [34], the cholera outbreak in Baghdad in 2007 [35], and many more.

However, since the 1960s, antibiotic resistance has gradually increased, while the number of new antibiotic classes brought to market has decreased. This will bring us back to the threat of epidemics and bacterial infections, which will have a long-term effect on the history of humanity and anthropology [36,37]. Unless the world responds quickly, antibacterial resistance will have a long-term negative effect on modern human history. Already, drug-resistant infections claim at least 700,000 lives each year worldwide, but if nothing is changed, that number may rise to 10 million deaths by 2050 [38,39,40]. This is precisely what occurred when superbug bacteria (extremely resistant to antibiotics) started to arise in many regions of the planet, signaling an impending global disaster that might threaten human society.

Therefore, the WHO has published a list of antibiotic-resistant bacteria of the utmost worldwide priority in order to drive the research, discovery, and development of new antibacterial drugs, the majority of which are Gram-negative bacteria (Table 1) [41]. Most of these bacteria were identified centuries ago and were kept under control throughout the antibiotic golden era, but they have recently become life-threatening globally.

## 5. The Post-Antibiotic Era

In 1945, Alexander Fleming, who had earned the Nobel Prize for his discovery of penicillin, cautioned that its abuse may lead to the spread of resistant bacteria. Within 10 years of the widespread adoption of penicillin, resistance started to arise, as predicted [42]. Since then, a race has started between bacterial infections and newly produced antibiotics. Significant progress was made with the discovery and development of sulfonamides, penicillin, and streptomycin. The characterization of tetracyclines, macrolides, cephalosporins, nalidixic acid, and glycopeptides followed these accomplishments [43,44,45].

After the golden era of antibiotics from 1930 to 1960, there was a period of stagnant innovation between the 1960s and the 2000s, during which antibiotic resistance grew. However, the advent of medicinal chemistry facilitated the rapid discovery of numerous classes of antibiotics with a variety of bactericidal or bacteriostatic modes of action. Regrettably, bacteria have acquired a multitude of resistance mechanisms in a similar manner (Figure 2). Indeed, beginning in the 1970s and continuing into the 2000s, the pharmaceutical industry’s declining desire and capacity to create new antibiotics culminated in a 40-year era during which essentially no new broad-spectrum antibiotic classes were introduced to the market. Instead, companies concentrated on changing the chemical scaffolds of already-approved antibiotic classes [46].

Undoubtedly, from 1928 to 1960, five major antibiotic classes were invented: penicillins in 1928 [48], tetracyclines in 1948 [49], macrolides in the 1950s [50], carbapenems in 1976 [51], and fluoroquinolones in the 1960s [52], followed by a lengthy innovation gap during which only one new class, teixobactin, was invented in 2015 [53]. Although teixobactin, as a new class of antibiotic, is similar in action to penicillins, it differs in its ability to bind to peptidoglycan’s lipids [54]. According to the current pipeline assessment, 43 novel antibiotics were under development as of December 2020 [55]. The majority these novel antibiotics do not possess a novel mechanism of action; rather, they are adaptations or modifications of existing antibiotic classes [56]. Many, but not all, resistant bacteria could be treated with these novel antibiotics. Given the likelihood that some of these antibiotics will fail to receive approval and also that resistance will evolve to those that do, it is evident that there are insufficient antibiotics in development to fulfill present and future patient demands [55,57,58]. Surprisingly, according to the WHO, 27 of the 43 antibiotics authorized by the FDA in 2020 are non-traditional antibacterial agents. These include nine antibodies, eight microbiome-modulating agents, four phage-derived enzymes and bacteriophages, two immunomodulatory agents, and four diversified antibacterial molecules [59]. This illustrates the new progress of antibacterial therapy, when scientists have begun to develop non-antibiotic antibacterial agents. However, the problem is still there: more than 70% of bacteria that cause serious diseases are likely to be resistant to at least one of the regularly used antibiotics [60]. Therefore, the rise of antibiotic resistance, along with the deficiency of novel antibiotics, presents a bleak picture of the future [61].

## 6. Antibiotics at a Crossroads: Unraveling the Missteps

At the beginning, many antimicrobial drugs, such as penicillin, were derived from natural sources such as fungi or bacteria. For example, vancomycin, teicoplanin, and daptomycin inhibit cell wall formation and come from *Amycolatopsis orientalis*, *Actinoplanes teichomyceticus*, and *Streptomyces roseosporus*, respectively. Other drugs, such as streptomycin, erythromycin, and gentamicin, inhibit protein synthesis and are obtained from microorganisms such as *Streptomyces griseus*, *Streptomyces erythreus*, and *Micromonospora* sp. Colistin and polymyxin B disrupt the permeability of the cytoplasmic membrane and have been isolated from *Bacillus colistinus* and *Bacillus polymyxa*. Lastly, carbapenem and cephalosporin drugs inhibit cell wall synthesis and are derived from *Streptomyces cattleya* and *Cephalosporium acremonium*, respectively [62,63]. As we discuss below, the excessive utilization of synthetically produced antibiotics has been linked to significant side effects and poses risks to the microorganisms involved in biogeochemical cycling. Moreover, the potential lethality associated with antibiotic-resistant bacteria is comparable to that of viral epidemics or pandemics. For instance, it was reported that the COVID-19 pandemic posed a substantial risk to hospitalized patients and immunocompromised individuals, resulting in an increased prevalence of secondary infections, often caused by antibiotic-resistant bacteria [64]. Antibiotic resistance is driven by two primary factors: bacterial ability and human influence:

### 6.1. The Bacterial Factor

Bacteria, being some of the most successful microorganisms on Earth, have demonstrated remarkable adaptability and resilience in various environments. Bacteria’s adaptability has allowed them to persist on the Earth’s surface for up to 2 billion years [65]. The discovery of novel antibacterial drugs and the development of new approaches demand a thorough understanding of the molecular mechanisms of antibiotic resistance. Recently, bacterial cells have been shown to be capable of developing a variety of resistance mechanisms, and the combination of these mechanisms may result in the emergence of multidrug-resistant pathogens or superbugs. As shown in Figure 3, bacteria may acquire resistance to any antibacterial drug via a variety of mechanisms, including the following:(1)Mutation: A spontaneous alteration in the DNA sequence of the gene may affect the trait for which it codes. A single base-pair change may alter the expression of one or more amino acids, thereby changing the enzyme or cell structure and resulting in resistance to the targeted antibiotic [66].(2)Plasmid-mediated resistance: The spread of antibiotic resistance genes that are on plasmids. The plasmids can be moved between bacteria of the same species or between bacteria of different species by conjugation. Plasmids often have a lot of different antibiotic resistance genes on them, which help spread multidrug-resistant (MDR) bacteria. Antibiotic resistance caused by MDR plasmids severely limits the treatment options for infections caused by Gram-negative bacteria [67].(3)Biofilm formation: Bacteria that stick to damaged tissue or transplanted medical devices often enclose themselves in a wet matrix of polysaccharides and peptides, forming a slimy coating called a biofilm. The biofilm’s resistance to antibiotics is dependent on complex multicellular mechanisms [68].(4)Quorum-sensing: A bacterial communication strategy that is dependent on the bacterial population density. It entails the use of tiny dissolved signaling molecules to stimulate the expression of a large number of genes that regulate a wide variety of activities, including antibiotic resistance [69]. This process has been clearly shown in the development of resistance when *Pseudomonas aeruginosa* moves to a new niche and is exposed to antibiotics [70].(5)Outer membrane permeability modification: The outer membrane of Gram-negative bacteria, in particular, acts as a strong barrier to antibacterial treatments. Antibiotics may pass through the outer membrane in two general ways: through a lipid-mediated pathway for hydrophobic antibiotics or through general diffusion porins for hydrophilic antibiotics. Bacteria are extremely efficient at using both of these mechanisms to enhance their resistance to antibiotics through modifications to these macromolecules [71].(6)Efflux pumps: Bacteria use this process to expel harmful solutes (e.g., antibiotics) from the cell. Antibiotic efflux in bacteria was first reported in the 1970s for tetracyclines. Since then, multidrug efflux pumps have received a lot of attention, and this pathway has been found in a variety of MDR bacterial species, particularly in Gram-negative bacteria [72].(7)Reduced uptake: Antibiotic-resistant bacteria reduce membrane permeability in one of two ways: by increasing the efflux or by decreasing the uptake. The reduced uptake (decreasing uptake) was shown to be responsible for beta-lactam resistance in Gram-negative bacteria, as beta-lactams need to penetrate the periplasmic region to bind the penicillin-binding protein targets located in the cytoplasmic membrane [71].(8)Inactivation of antibiotic by bacterial enzymes: Clinically resistant bacteria express modifying enzymes that inactivate the antibiotics; these cellular enzymes are changed to react with the antibiotic in such a manner that it no longer affects the microorganism [73,74].(9)Alternation of the antibiotic target: Antibiotic target-site alteration is a frequent mechanism of resistance; it occurs to evade the antibiotic’s effect by interfering with its target site. To do this, bacteria have developed a variety of strategies, including target protection (preventing the antibiotic from reaching its receptor) and changes via mutation of the target site, resulting in a lower sensitivity to the drug [75].

### 6.2. The Human Factor

Antibiotic consumption patterns, the misuse and overuse of antibiotics, patient demands, and human behaviors regarding antibiotics are widely acknowledged as significant contributors to the issue of antibiotic misuse and overuse [76]. Self-medication is the most prevalent cause of human pathogen resistance to antibiotics, and it was discovered that 20% of human antibiotic usage occurred in hospitals, while 80% occurred in communities. Up to 50% of community usage is dubious and needless [77,78]. The prevalence of nosocomial infections and antibiotic misuse are significant public health concerns, since antibiotics are commonly used in hospitals for surgical treatment and nosocomial infections, which appear more commonly in 15–20% of hospitalized patients [79,80]. Regrettably, humans utilize antibiotics extensively in food animals to stimulate growth and prevent illness, and multiple studies have demonstrated that antibacterial resistance resulting from antibiotic usage in food animals has a detrimental effect on human health and the development of resistant bacteria [81,82]. A similar phenomenon has been found in aquaculture, where the widespread use of antibiotics in large quantities in fish food raises the possibility of residual antibiotics being present in fish flesh and fish products [83]. Moreover, antibiotic residues from pharmaceutical companies and the accumulation of massive amounts of discarded antibiotics in the environment pose a severe environmental danger, since the global antibiotic market uses between 100,000 and 200,000 tons annually [84,85]. In addition, antibiotic residues in the environment may have a number of negative ecological consequences. Climate change may be a factor in the pathogen spread risk. Changing climatic factors have the potential to modify the distribution and density of organisms, resulting in new interactions between them and increased chances of infectious diseases emerging [86]. Therefore, human behavior in the mass manufacture of antibiotics, residue deposition in nature, and antibiotic misuse in medicine, veterinary medicine, and agriculture have all contributed significantly to the rise of the phenomenon of antibiotic-resistant bacteria (Figure 4).

## 7. Emerging Global Focus on Therapeutic Applications of Medicinal Plants

Traditional medicinal plants were used as treatments long before the establishment of western medicine with the emergence of modern science and technology [1,87]. Plant-derived antibacterial compounds were the bases of antimicrobial drugs before the antibiotic era [88]. The number of plant species (*Angiosperms*) on Earth is believed to be between 250,000 and 500,000. A very small number of these plants (less than 10%) are eaten by humans and other animals, and many are used for medical purposes [87]. Despite a temporary decline during the Industrial Revolution, popular interest in medicinal plants persists globally. The WHO reports a significant and escalating global trend in the utilization of herbal medicines and phytonutrients for addressing diverse health concerns within varied national healthcare systems [89]. Although plants are the primary healthcare source for 80% of individuals in developing countries, herbal remedies have gained acceptance in developed nations, where complementary and alternative medicines have become mainstream, particularly in the UK, Europe, North America, and Australia [90]. In response to the widespread emergence of antibiotic-resistant bacteria, the investigation and application of naturally derived compounds from medicinal plants have garnered substantial attention among researchers [91]. In the present review, a comprehensive search was carried out on scientific reports published in Google Scholar, focusing on studies investigating the antibacterial and antimicrobial properties of medicinal plants between 2010 and 2022. Our survey uncovered a notable surge of interest among the global scientific community in this domain, implying the potential emergence of novel antibacterial agents derived from certain plant sources in the near future (Figure 5).

## 8. The Mechanisms of Plant-Derived Antibacterial Agents

Following the scientific validation of the antibacterial efficacy of numerous medicinal plants (Figure 5), the subsequent step in the utilization of medicinal plant molecules as natural antibacterial agents entails a comprehensive understanding of the underlying mechanisms dictating their mode of action. Generally, the antibacterial properties of medicinal plants are hypothesized to primarily stem from two mechanisms: chemical interference with the synthesis or functioning of crucial bacterial components and/or bypassing the conventional mechanisms of antibacterial resistance [92]. The illustrated mechanism corresponds to established antibacterial medicinal plants, as depicted in Figure 6. Plants may restore the physiological balance of the body to make it more resistant to pathogens, and this philosophy is totally absent in antibiotic treatment (holistic mechanism). In modern medicine, drugs are mostly modified and synthesized in the form of single bioactive compounds to target a specific disorder or infection, whereas synergism is the general feature of traditional medicine, which provides multiple targets against specific diseases [93]. Medicinal plants, including garlic (*Allium sativum*), ginger (*Zingiber officinale*), green tea (*Camellia sinensis*), St. John’s wort (*Hypericum perforatum*), black cumin (*Nigella sativa*), licorice (*Glycyrrhiza glabra*), Mongolian milkvetch (*Astragalus membranaceus*), and purple coneflower (*Echinacea* spp.), possess a notable history of efficacy in managing microbial diseases. These plants exhibit noticeable immune-boosting properties and have the potential to combat bacterial pathogens, provided that they are thoroughly researched and effectively utilized [94]. The predominant antibacterial mechanism of action exhibited by essential oils derived from polyphenol- and terpene-rich plants is the disruption of the membrane function and the structure of bacterial cells such as *Cuminum cyminum*, *Mentha piperita*, *Thymus daenensis*, *Pimenta dioica*, *Myrtus communis*, and others, involved the disruption of the plasma membrane [95]. Essential oils, particularly those derived from the *Lamiaceae* and *Verbenaceae* families commonly found in the Mediterranean region, exhibit anti-quorum sensing and anti-biofilm properties against bacterial pathogens [96]. Moreover, secondary metabolites can exert various effects on microbial cells, including interference with intermediary metabolism, the disruption of DNA/RNA synthesis and functionality, and the modulation of critical events within the pathogenic progression [97].

## 9. Major Phytochemical Classes with Potent Antibacterial Activity

Plants have two major groups of metabolites: primary and secondary. Carbohydrates and lipids are products of the primary metabolism of plants, while phenolic compounds, carotenoids, alkaloids, saponins, and terpenoids are considered to be secondary metabolites. Numerous secondary metabolites exhibit multifaceted pharmacological properties, such as anti-inflammatory, antitumor, antioxidant, and antimicrobial activities, among others [92]. Next, an elaborate discussion concerning these substances and their significance as potential reservoirs of molecules with notable antimicrobial effects will subsequently be delved into (Appendix A).

### 9.1. Phenolic Compounds

Known for providing protection to plants against microbial agents, oxidants, and ultraviolet radiation, phenolic compounds are characterized by the presence of a benzoic ring in their chemical structure [98]. Currently, more than 8000 phenolic compounds with some bioactivity have been categorized [99], including phenolic acids and aldehydes, flavonoids, chalcones, benzophenones, xanthones, stilbenes, benzoquinones, and polyphenols, among others, which can be extracted from different parts of the plant, such as the leaves, roots, and fruits (bark and seeds) [100]. Studies indicate that these compounds are more effective against Gram-positive bacteria, which can be explained by the presence of the thick layer of peptidoglycan and the absence of an external membrane found in Gram-negative bacteria, which exert a hydrophobic action, preventing the penetration of hydrophilic molecules into the bacterial cell, such as phenolic compounds [98]. The main mechanism of action of phenolic compounds is related to their ability to reduce the expression of efflux pumps [101]. However, there are reports of molecules from this group that inhibit DNA gyrase and, thus, are capable of inhibiting microbial growth, as is the case for tannins [102] and anthraquinones [103]. Poomanee et al. [104] found promising results of the antimicrobial action of phenolic compounds extracted from the seeds of *Mangifera indica* against standard strains of *Staphylococcus aureus* and *Staphylococcus epidermidis*. Similarly, Lyu et al. [105] found that phenolic compounds from extracts of *Hibiscus acetosella* inhibited the growth of *S. aureus*, in addition to being effective in the microbial control of *P. aeruginosa*. Baicalein, a flavonoid isolated from the roots of *Thymus vulgaris*, *Scutellaria baicalensis*, and *Scutellaria lateriflora*, has shown good antimicrobial activity against viral and bacterial isolates [106]. Another study demonstrated that the compound baicalein increased the susceptibility of methicillin-resistant *S. aureus* (MRSA) to beta-lactams, tetracycline, and ciprofloxacin through the inhibition of efflux pumps [107]. Within this group, it is still worth highlighting the chalcones, which can impede microbial growth by inhibiting the expression of efflux pumps. This is the case of the 4′,6′-dihydroxy-3′,5′-dimethyl-2′-methoxylic chalcone isolated from *Dalea versicolor*, which showed good antimicrobial action against *S. aureus*, being able to inhibit the expression of the *NorA* efflux pump [108].

### 9.2. Alkaloids

The term alkaloid means “similar to alkalis”, referring to the basic or alkaline character of the substances. About 12,000 alkaloid compounds isolated from plant extracts have already been categorized, with various medicinal actions, such as antitumor, analgesic (morphine and codeine), and antimicrobial properties [109]. They present a chemical structure with heterocyclic rings containing N-heterocyclic nitrogen and can be classified according to their carbon precursors and structure [98]. Examples of alkaloid compounds commonly found in plants include pyridine, piperidine, quinoline, alkaloidal amines, and terpenoid [110]. The antibacterial action of the alkaloid fractions isolated from *Callistemon citrinus* leaves was demonstrated against Gram-positive and Gram-negative bacterial isolates, such as *S. aureus* and *P. aeruginosa*, by Mabhiza et al. [111]. The authors believe that the compound acted by inhibiting the transport of ATP-dependent substances across the bacterial cell membrane. Liu et al. [112] identified the good antimicrobial action of benzyltetrahydroisoquinolin-derived alkaloids from the leaves of *Doryphora aromatica* against isolates of *Mycobacteria* spp. and *S. aureus* resistant to methicillin. Pech-Puch et al. [113] verified the good (MIC in the range of 1–8 µg/mL) and moderate (MIC value of 16 µg/mL) antimicrobial action of diterpene alkaloids from *Agelas citrina* against the Gram-positive pathogens *S. aureus*, *S. pneumoniae*, and *E. faecalis* and the Gram-negative pathogens *A. baumannii*, *P. aeruginosa*, and *K. pneumoniae*,. Reserpine is an indole alkaloid isolated from the *Rauwolfia serpentinaa* species of flower in the Apocynaceae family, native to the Asian continent, which has been shown to increase the susceptibility of multidrug-resistant clinical isolates of *A. baumannii* and *S. maltophilia* [114,115].

### 9.3. Saponins

Saponins can be found in a variety of plants and are chemically characterized by the presence of glycosylated groups, formed by a hydrophilic and a lipophilic part. This structure confers the properties of detergents and surfactants on saponins [92,98]. One of the reported properties of saponins is antimicrobial activity. It is known that the chemical structure of these compounds directly interferes with the effectiveness of their antimicrobial action. There have been reports that saponins with trisaccharide chains exhibited good antifungal action, whereas saponins with mono- or disaccharide chains did not show good antimicrobial action [116]. Lunga et al. [117] observed the promising antimicrobial action of different saponins from *Paullinia pinnata* against Gram-positive and Gram-negative bacteria, as well as yeast. From *Cephalaria ambrosioides*, some triterpenoid saponins, cauloside A, α-hederin, dipsacoside B, and sapindoside B were isolated, showing strong anti-bacterial activity against *S. aureus*, *S. epidermidis*, *P. aeruginosa*, *E. coli*, *E. cloacae*, and *K. pneumoniae* (MIC values 1.80–2.50 µg/mL) [118].

### 9.4. Terpenoids

Terpenoids, or terpenes, are a class of metabolites that encompass a variety of natural substances, which have in common the presence of C_5_ isoprene units in their chemical structure. Depending on the amount of C_5_ isoprene involved in their synthesis, terpenes can be classified as monoterpenoids, sesquiterpenoids, diterpenoids, sesterterpenoids, and triterpenoids. More than 40,000 terpenoid substances are known, with different applications: aromatic, pharmaceutical, agricultural, and industrial [98]. In addition, there are reports of the promising antimicrobial action of different types of terpenes. For example, Biva et al. [119] demonstrated the antibacterial action of terpene compounds from *Eremophila lucida* against *S. aureus* isolates. Similarly, Gartika et al. [120] observed that terpenoids obtained from *Myrmecodia pendans* showed promising antibacterial action against *S. mutans*—an important caries-related pathogen—in addition to being effective in inhibiting and eradicating *S. mutans* biofilm. Zhu et al. [121] isolated and identified terpenoids from *Commiphora* resin, with good antibacterial action against sensitive and resistant isolates of *Mycobacterium tuberculosis*.

### 9.5. Other Compounds

Many phytochemicals not mentioned above have been found to exert antimicrobial properties, including mucilage, essential oils, fixed oils, sterols, and waxes (Appendix A). Lipids are a class of naturally occurring compounds that include essential oils, fixed oils, sterols, waxes, phospholipids, and fat-soluble vitamins. They were once categorized as primary metabolites, but studies have shown them to have secondary metabolite functions [122]. While carbohydrates are classified as primary metabolites, specific carbohydrates have been discovered to have functional properties and are categorized as secondary metabolites, such as mucilage, which is produced as a phytochemical by a variety of plants and exhibits a wide range of biological activities [123]. Some secondary metabolites, on the other hand, are made mostly by regulators that are activated by certain carbohydrates [124]. Moreover, a published report showed the antibacterial activity present in garlic and onions, which exhibit inhibitory effects on diverse microorganisms due to their abundant sulfoxide contents, which impart them with antimicrobial properties. On the other hand, the horseradish, mustard seeds, and wasabi demonstrate inhibitory activity, attributed to their elevated levels of allyl glucosinolates [125].

## 10. Medicinal Plants versus Antibiotics

Utilizing medicinal plants in the innovation of new antibacterial drugs offers numerous advantages in comparison to synthetic or semisynthetic antibiotics. These include the following:(i)Medicinal plants offer advantages over conventional antibiotics in terms of availability and cost-effectiveness. They are easily accessible and more economical compared to large-scale antibiotic production, as they do not require extensive and expensive chemical and pharmaceutical procedures. In contrast, synthesizing new antibiotics is a complex and time-consuming process that is prone to setbacks and high costs. Developing a novel antibiotic typically demands significant resources, taking 10 to 15 years and costing over USD one billion [126,127].(ii)Herbal medicines interact safely with the body’s vital systems, exhibiting minimal side effects. They are efficiently eliminated through the excretory system and often have synergistic effects that promote physiological balance. In contrast, many antibiotics are either semisynthetic derivatives/chemically synthesized, with potential negative side effects and a risk of contributing to antibiotic resistance with frequent use [128,129].(iii)Medicinal plants possess remarkable potential for generating a wide array of bioactive molecules with antibacterial properties, offering a vast and inexhaustible repertoire. In contrast, the sources of antibiotics are considerably limited, and the global supply is dwindling [1,130](iv)Medicinal plant products pose minimal pollution risks and can be extracted using eco-friendly methods. In contrast, antibiotics necessitate reduced usage due to their negative effects on soil and water pollution. The annual manufacturing and residue discharge of antibiotics contribute significantly to a pollution load estimated between 100,000 and 200,000 tons [131,132].(v)Medicinal plants exhibit multiple complementary and synergistic mechanisms of action, rendering them highly promising for addressing antibiotic-resistant bacteria. In contrast, pathogenic bacteria have developed diverse mechanisms and strategies to significantly evade the effectiveness of antibiotics that rely on a single molecule [133,134].

On the other hand, three critical factors need to be considered for the production of any successful antibacterial drug: the comparison of the efficacy between the new product and the old synthesized antibiotic; the assurance of product safety and the reduced occurrence of complications, side effects, or toxicity; and the motivation of pharmaceutical companies to engage in its production. A detailed description of these three factors is provided below.

### 10.1. Safety of Antibacterial Phytochemicals

Numerous side effects have been reported due to the misuse, overdose, or even long-term treatment of antibiotics, including cardiovascular depression, respiratory difficulties, or alterations of the metabolic breakdown of other drugs.

Although the use of herbal and natural medicines has fewer side effects compared to conventional drugs, knowledge about the mechanisms of action, possible drug interactions and their consequences, bioavailability, and effective dosage and time required for treatment is still scarce. This is because, although the technologies used in the research of new drugs have evolved in recent decades, the process of the discovery, isolation, and evaluation of natural compounds as possible medicinal agents is still slow-paced, wastes a lot of time, and is expensive and complex. In addition, we observed that the vast majority of compounds that show good action in vitro are no longer eligible for clinical trials in vivo, due to either the observation of their toxicity or their low effectiveness [135].

### 10.2. Effectiveness of Plant-Based Compounds

Several studies have demonstrated the effectiveness of natural compounds, especially in in vitro assays, in inhibiting the growth and even the production of virulence factors by different types of microorganisms, including bacteria, fungi, and parasites. For example, Menezes et al. [135] demonstrated the effectiveness of the capsaicin molecule, found in peppers of the genus *Capsicum*, in controlling the growth of *Candida* spp. and *Toxoplasma gondii*, in addition to its ability to inhibit the formation of biofilm by *Candida* spp. It has been reported that the antimicrobial action of the compound piperine, isolated from another group of peppers, was effective against *S. aureus* and *Bacillus subtilis* at low concentrations by inhibiting efflux pumps. Fu et al. (2021) observed in their study that the alkaloid was able to inhibit biofilm formation in 90% of isolates of *Serratia marcescens* that were resistant to carbapenems from 32 µg/mL [92,136]. Despite the good results shown in the in vitro tests, some authors recognize the need to perform in vivo tests to assess the toxicity of compounds and prove their effectiveness in controlling infections [135,137].

### 10.3. Pharmaceutical Company Contentment

There are five major reasons that might drive pharmaceutical companies to invest in medicinal plants and natural products for the innovation of novel antimicrobial drugs:(i)The high cost of synthesizing novel antibacterial chemical compounds compared to the low cost of manufacturing antibacterial agents from natural products [138].(ii)Large pharmaceutical corporations have departed the market of synthetic antibiotics due to a lack of financial incentives and profits [139].(iii)Synthetic antibiotics have not been able to stop the spread of bacterial pathogens that have become highly resistant [140].(iv)The abundance of phytochemical molecules isolated from medicinal plants that are powerful against bacterial infections and have been scientifically confirmed [87].(v)New advances in biotechnology make it possible to manufacture novel antibacterial drugs from plants with great efficiency [1].

## 11. Plants Exhibiting Antibacterial Potential against WHO-Designated High-Priority Pathogens

Regarding recent research on plant extracts’ antibacterial action, we carried out a survey of the PubMed databases (which include Web of Science, Scopus, and SciELO, among others), with the objective of identifying publications that evaluated the antibacterial action of plant extracts against species considered to be a critical priority by the WHO. For this, Medical Subject Headings (MeSH) descriptors were used through the Boolean operator and listed from a search in the Lilacs database, which are listed as follows: antimicrobial natural compounds, antibacterial natural compounds, plant extracts, medicinal plants, *Acinetobacter baumannii*, *Escherichia coli*, *Enterobacter* spp., *Klebsiella pneumoniae*, *Pseudomonas aeruginosa*, *Proteus* spp., *Serratia* spp., *Enterobacteriaceae*, carbapenem-resistant, and third-generation-cephalosporin-resistant. The descriptors were used in English to carry out the bibliographic survey. The articles resulting from the search underwent a preliminary evaluation, with titles and abstracts being analyzed in order to verify whether the study addressed the theme proposed in this review. In studies in which the reading of the title and abstract was not sufficient for the application of the inclusion and exclusion criteria, the publication was read in full. This review included articles published between 2018 and 2022 in English or Spanish, with results of experimental research on the antimicrobial action of plant extracts against pathogens considered to be of critical importance by the WHO, and which could be accessed in full in the databases used. Likewise, duplicate articles, those without open access, reviews, those that evaluated the action of isolated molecules acquired commercially, and those that noted the plant of origin of the evaluated extract were excluded from this research.

The results of the survey of articles in the selected databases are shown in the Appendix A. In total, the search, combined with the listed descriptors, resulted in 504 articles, of which 11 met the applied inclusion and exclusion criteria, as listed in Appendix A, along with another 12 plant species cited in the last 4 years as potent antibacterial agents against some pathogens enlisted in the WHO priority list. Most articles evaluated the antimicrobial action of aqueous extracts and hexane. The most frequent isolate in the studies was carbapenem-resistant *P. aeruginosa* (72.7% of studies), followed by carbapenem-resistant *A. baumannii* and carbapenem-resistant *K. pneumoniae* (54.5% of studies). The lowest MIC values (100 μg/mL) were found by Nocedo-Mena et al. [141] when evaluating the action of compounds isolated from *Cissus incisa* leaves against carbapenem-resistant *P. aeruginosa* (Table 2).

## 12. Synergistic Interactions of Phytochemicals against Bacterial Pathogens

Synergism is an intriguing mechanism by which plant-derived compounds manifest their antibacterial efficacy. Currently, scientists are pursuing three main approaches to capitalize on the distinctive synergistic properties exhibited by antibacterial molecules derived from plants. These strategies include (i) plant molecules’ synergistic interactions with antibiotics, (ii) synergistic combinations of bioactive plant molecules, and (iii) plant molecules’ synergistic interactions with nanomaterials. Subsequently, a detailed explanation will follow.

### 12.1. The Synergistic Activity of Plant Molecules with Antibiotics

A substantial body of research has consistently demonstrated that the supplementation of plant extracts with antibiotics has the potential to enhance their efficacy, concurrently diminishing both the required dosages and any associated adverse effects. These advantageous interactions are regarded as a promising strategy in the ongoing battle against bacterial resistance [164]. As brief examples, the ethanolic leaf extract of *Plectranthus ornatus* demonstrated a synergistic effect when combined with ampicillin (a β-lactam antibiotic), kanamycin, and gentamicin (both aminoglycoside antibiotics), resulting in an eightfold decrease in the MIC values against various strains of *Staphylococcus aureus* [165]. Acetone, chloroform, ethyl acetate, and methanol extracts of *Helichrysum longifolium* were combined with six antibiotics, showing significant synergistic effects against bacterial isolates and enhancing the bactericidal effects of the antibiotics according to the time-kill and checkerboard methods [166].

In a study exploring the synergistic interaction between constituents of *Punica granatum* and various antibiotics, the authors suggested that the methanol extract of the plant contributes to efflux inhibition, consequently augmenting the uptake of the antibiotics [167].

### 12.2. Synergistic Combinations of Bioactive Plant Molecules

Numerous phytochemical molecules of plants manifest their advantageous impacts by virtue of the additive or synergistic action of multiple phytochemical compounds, which act upon single or multiple target sites, offering new possibilities for the development of antibacterial interventions [168]. The combination of essential oils derived from the aerial parts of *Thymus vulgaris* and methanol extracts of *Pimpinella anisum* seeds exhibited an additive effect against a range of tested pathogens, including *Staphylococcus aureus, Bacillus cereus, Escherichia coli, Proteus vulgaris, Proteus mirabilis, Salmonella typhi, Salmonella typhimurium, Klebsiella pneumoniae,* and *Pseudomonas aeruginosa* [169].

During the assessment of the antibacterial properties of *Eucalyptus globulus*, *Cymbopogon martinii*, *Cymbopogon citratus*, and *Mentha piperita*, it was observed that, in the majority of cases, the MIC values of the whole oils were lower compared to the MIC values of the primary individual constituents present in those oils. This finding suggests the presence of synergistic interactions, resulting in increased and enhanced antibacterial efficacy [170]. Despite the potential for synergistic combinations between plant extracts, it has been observed that such synergies are not always successful. A study investigating combination trials between essential oils from 10 commonly used spices and herbs reported that only the coriander/cumin seed oil combination exhibited notable and significant synergistic interactions in terms of antibacterial activity, and this underscores the complexity and variability in the synergistic potential of plant extract combinations [171].

### 12.3. The Synergistic Activity of Plant Molecules with Nanomaterials

Plant molecules can also exhibit synergistic effects when combined with nanomaterials. The combination of plant molecules and nanomaterials shows enhanced antimicrobial properties, which can be utilized for the development of new antimicrobial therapies [172]. Scientists have endeavored to utilize non-toxic nanomaterials to create virus-sized nanoparticles capable of encapsulating pharmaceuticals. These nanoparticles serve as transport vehicles for antibacterial drugs, effectively fusing with the bacterial membrane. This enables the targeted delivery of the antibacterial agent directly to the bacterial cell, resulting in the eradication of the bacteria [173]. The green synthesis of silver nanoparticles (AgNPs) incorporated with aqueous extracts of *Thymus vulgaris*, *Mentha piperita*, and *Zingiber officinale* exhibited significant antibacterial activity against three multidrug-resistant clinical isolates, namely, *Escherichia coli*, *Acinetobacter baumannii*, and *Staphylococcus aureus* [174]. The green synthesis of zinc oxide nanoparticles with *Withania somnifera* leaf extract demonstrated enhanced antibacterial activity against *Enterococcus faecalis* and *Staphylococcus aureus* at a concentration of 100 µg/mL. Additionally, these nanoparticles exhibited a significant inhibition of the biofilm formation by *E. faecalis* and *S. aureus* at the same concentration [175]. Another study revealed that the *Laurus nobilis* leaf-extract-mediated synthesis of zinc oxide nanoparticles has selective antibacterial effects, as it exhibited a higher efficacy against Gram-positive *Staphylococcus aureus* compared to Gram-negative *Pseudomonas aeruginosa*, potentially due to variations in the structural composition of the bacterial cell walls [176].

## 13. Challenges in the Field of Plant-Based Drug Discovery

According to the above, and based on the literature, we observed that there are a multitude of substances in nature with antibacterial potential that have shown effective action in isolation or in combination with conventional antibiotics. These findings point out directions to follow in the search for alternatives for combatting pathogens that are considered to be a high priority by the WHO [177]. Although the literature points to promising results in the use of natural compounds as antibacterial agents for in vitro assays, there is a long way to go until some of these compounds actually become drugs, as little is known about their side effects and mechanisms of action. So, more in vitro and in vivo studies are needed to figure out the mechanism of action, bioavailability, safety, effectiveness, bacteriostatic or bactericidal activity, rate of resistance, bioavailability, and stability of the compounds in the formulation, along with the pharmacokinetic studies (in vivo), spectra of antibacterial activity, methods of administration, treatment duration, and possible interactions with bodily fluids and tissues in the short term and the long term [178]. Moreover, there is a lack of standardized methodologies for in vitro testing, and there are no set cutoffs to help with the correct interpretation of data [179]. It was observed that the majority of the published studies are academic publications without any intention to go deeper to isolate an active antibacterial principle or to innovate a new drug. This is because the development of new drugs, even from natural products, requires great efforts, a long time, high costs, and complex studies that only huge pharmaceutical companies or international cooperation between research centers can carry out.

## 14. New Perspectives on Antibacterial Agents

In recent years, the development of new antibacterial agents has become crucial to combatting the emergence of resistant pathogens. The utilization of medicinal plants in their traditional state as antibacterials is limited due to the influence of chemical constituents on seasonal variations and the climate [180].

Presently, researchers worldwide are exerting efforts to advance the development of antibacterial agents or identify viable alternatives from many sources, including plant molecules, in order to mitigate potential consequences before reaching a critical stage. Regarding plant-based antibacterials, green chemistry and nanotechnology could be an effective tool for increasing the efficiency of antibacterial agents derived from medicinal plants [181]. In the realm of herbal drug utilization, the emerging field of in silico drug discovery presents a novel approach. In particular, in silico high-throughput screening (HTS) employing molecular docking techniques is extensively employed to streamline the selection of compounds for subsequent in vitro and in vivo screening [97]. Consequently, HTS serves as a potent tool that is capable of rapidly screening a vast array of natural compounds within a condensed timeframe, facilitating the identification of potential therapeutic agents. Interestingly, the utilization of nanotechnological approaches has emerged as a new trend in plant-based drug discovery, wherein phytochemicals are subjected to nanonization via encapsulation or entrapment within hydrophilic capping agents of either an inorganic or organic nature. This nanonization process not only confers water solubility to these products but also enables the achievement of a significantly elevated surface-to-volume ratio. Consequently, the nanonized products exhibit enhanced therapeutic potential, surpassing that of their raw bulk counterparts when compared on an equivalent dosage basis [182]. Another brand-new perspective on plant-based antibacterial discovery is the use of artificial intelligence (AI); by utilizing an AI algorithm, researchers employed a screening process to analyze numerous antibacterial molecules documented in published reports. Their aim was to anticipate novel structural classes of antibacterial agents derived from potent plants. As a consequence of this AI screening, scientists successfully recognized a fresh antibacterial compound and swiftly examined extensive chemical territories, substantially augmenting the probability of unearthing fundamentally innovative antibacterial substances [183,184]. AI tools such as LeafNet, ANtiSMASH, and AutoDock aid in taxonomic identification, genomic analysis, and target identification for biosynthesis and compound design [185]. On the other hand, the benefits of utilizing liposomes as antibiotic carriers in nanomedicine vary from reduced toxicity to improved pharmacokinetic characteristics, particularly in terms of bio-distribution. The liposomal vesicles are fused with the bacterium’s surface membrane, which improves antibiotic delivery and makes it easier for the drug to get inside the bacterium [186]. Therefore, it may be possible in the future to replace antibiotics in the liposomal vesicles with natural antibacterial molecules extracted from medicinal plants and to test their effectiveness using this innovative technique, in conjunction with the utilization of advanced biotechnologies such as genomic approaches and proteomic methods. This study also recommends the standardization of in-house methodologies and developing the regular methodologies of investigating the antibacterial activities of medicinal plants by harnessing advanced technologies, such as high-throughput screening and bioinformatics, enabling researchers to screen thousands of plant extracts and identify novel molecules with antibacterial activity. Moreover, the emphasis must be placed on the return to Mother Nature and the utilization of its smart weapons against pathogenic bacteria, thereby shifting the focus from mere observation to gaining profound understanding.

## 15. Future Directions

The future directions of herbal drugs research entail a comprehensive investigation into the chemical composition and mechanisms of action of these plant-derived compounds. By pursuing these future directions, the scientific community can unlock the full potential of medicinal plants as a valuable resource in the fight against bacterial infections. Several of them can be succinctly summarized in terms of the following key aspects:(i)AI-driven precision medicine augments health-related tasks, providing highly personalized diagnostic and therapeutic information. Tailoring antibacterial treatments using medicinal plants to an individual’s genetics, bacterial profile, and health conditions maximizes efficacy and minimizes antibiotic resistance. This approach aims to prevent infections, reduce the disease burden, and lower healthcare costs for all [187].(ii)Modernization and Integration of Traditional and Modern Medicine: In global healthcare and infection control, more recognition and respect will be gained by traditional medicine and indigenous knowledge. The integration of traditional herbal remedies into evidence-based medical practices will be facilitated by collaboration between medicinal plants and modern pharmacoepidemiology. Such integration is already being applied in some countries, such as China and India [188].(iii)Rise of Plant Biotechnology: The enhancement of the antibacterial properties of medicinal plants will be crucially influenced by biotechnology and bioinformatics. Metabolic Engineering Strategies will be utilized to enhance the production of bioactive compounds, making them more potent and effective against drug-resistant bacteria [189].(iv)Combination Therapies: A more prevalent approach will involve the use of medicinal plant combinations with synergistic antibacterial effects or with conventional antibiotics. Specific herbal mixtures that work together to combat bacterial infections more effectively than single compounds will be the focus of researchers [190].(v)Standardization and Quality Control: Significant efforts will be made to standardize the production and quality control of herbal drugs to ensure their safety and efficacy as antibacterial agents. Regulations and guidelines will be put in place to maintain consistency across different formulations [191].(vi)Alternative Delivery Systems: Innovative delivery systems, such as nanoencapsulation and targeted drug delivery will be employed to enhance the bioavailability and targeted action of medicinal plant compounds against bacterial infections [192].(vii)Regulatory Support and Incentives for Global Collaboration and Research Sharing: International collaboration among researchers, governments, and pharmaceutical companies will be considered essential for advancing medicinal plant research. Open-access databases will be instrumental in facilitating the sharing of knowledge and data to accelerate drug discovery [193].

## 16. Conclusions

Infectious diseases, particularly those caused by bacterial infections, remain a prominent global cause of mortality. The rise of antibiotic resistance poses a grave threat to public health worldwide, as it leads to the proliferation of superbugs that are impervious to existing antibiotics. Experts have warned of an impending end to the antibiotic era. Unfortunately, it has been approximately four decades since the discovery of the last new class of antibiotic. Most antibiotics developed since then have been modifications of previously discovered classes from the golden era of antibiotics. This circumstance implies that resistance to these modified antibiotics may emerge rapidly, especially considering the accumulation of synthetic antibiotic waste in our environment over the decades. Furthermore, the pharmaceutical industry’s diminished interest in the antibiotics sector has exacerbated the challenge of identifying and producing novel antibiotics. Consequently, investments in antibiotic research have declined. The scientific community has increasingly recognized the futility of pursuing the synthesis of new antibiotics while repeating past mistakes. As a result, non-traditional antibacterial agents such as antibodies, phage-derived enzymes, immunomodulatory agents, and anti-virulence agents have gained attention in the clinical pipeline. Medicinal plants hold promise as a potential source for new antibacterial drugs due to their abundance of bioactive phytochemical compounds. Although numerous studies have reported on the efficacy of medicinal plants and their antibacterial properties, some analysts speculate that pharmaceutical companies may hesitate to invest resources in evaluating botanical drugs. This hesitation stems primarily from the prevailing uncertainty surrounding the ability to make exclusive claims regarding these drugs. This review presents compelling evidence supporting the antibacterial properties of various medicinal plants and their active compounds as promising alternatives. Many medicinal plants exhibit significant efficacy in targeting bacterial virulence factors and have potential for synergy with conventional antibiotics, making them viable candidates for further research and drug development. However, comprehensive investigations will be necessary to fully understand the mechanisms of action, toxicity, and pharmacokinetics of these plant-derived compounds. Nevertheless, the findings of this review establish a foundation for the development of novel and effective plant-based antibacterial drugs that can address the urgent public health threat posed by antibiotic-resistant bacteria. In line with regulatory practices for human medicines, it is now imperative for antibacterial herbal medicines to be encompassed within a comprehensive drug regulatory framework in all countries worldwide. Undoubtedly, there is a crucial imperative for effective coordination and collaboration among prominent entities such as the WHO, the FDA, the EMA, biotechnology companies, the pharmaceutical industry, and numerous regulatory agencies worldwide. This collaborative effort should aim to establish comprehensive and unambiguous guidelines for the exploration and advancement of plant-based antibacterial drugs, harnessing the extensive potential of traditional medicine in the development of therapeutic interventions for diverse and difficult-to-treat bacterial diseases.

## Figures and Tables

**Figure 1 plants-12-03077-f001:**
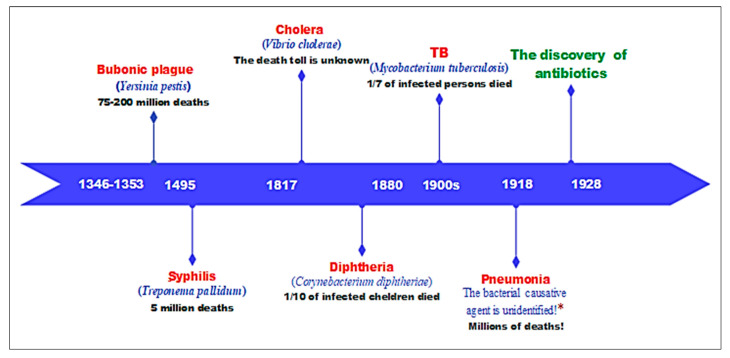
A timeline of epidemics from the early modern period onwards caused by bacteria prior to the discovery of antibiotics (***** during the Spanish flu pandemic, one or two of the following species were responsible for the majority of fatal pneumonia: *Streptococcus pneumoniae*, *Streptococcus pyogenes*, and/or *Staphylococcus aureus* [19]).

**Figure 2 plants-12-03077-f002:**
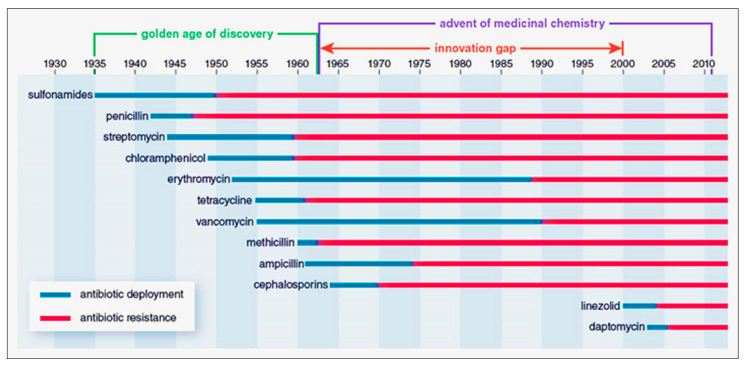
The emergence of antibiotics and the development of resistance to them (1935–2010) [46,47].

**Figure 3 plants-12-03077-f003:**
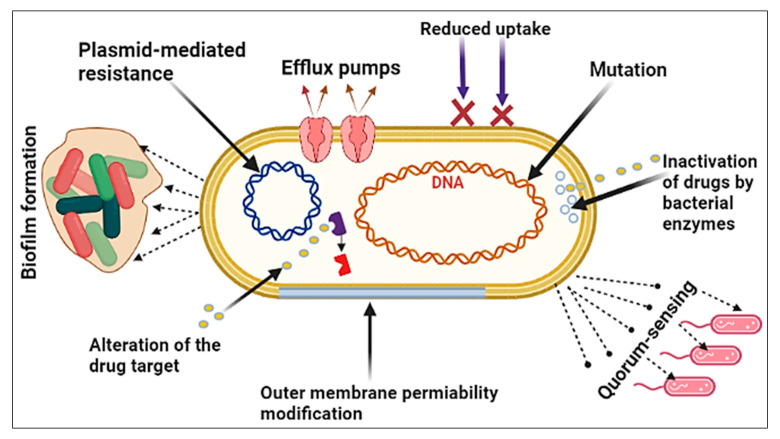
Major bacterial resistance mechanisms against antibiotics.

**Figure 4 plants-12-03077-f004:**
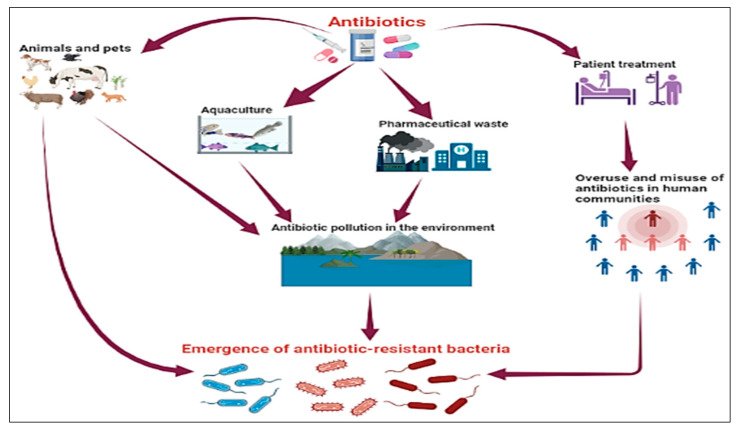
Significant human behaviors that have an impact on the spread of antibiotic-resistant bacteria.

**Figure 5 plants-12-03077-f005:**
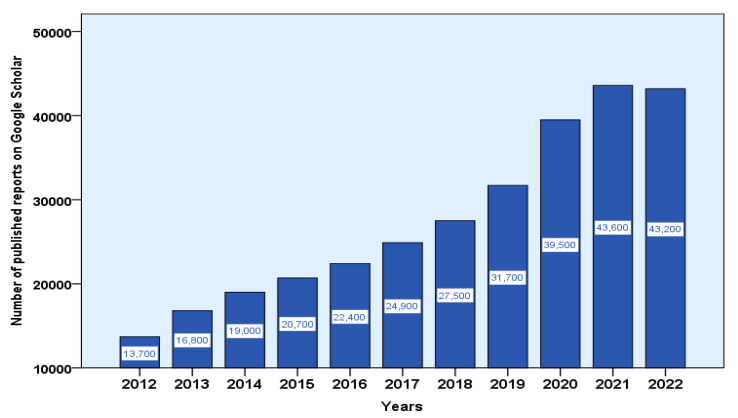
Growing global interest in the antibacterial properties of medicinal plants through the noticeable increase in the number of published reports over the past decade.

**Figure 6 plants-12-03077-f006:**
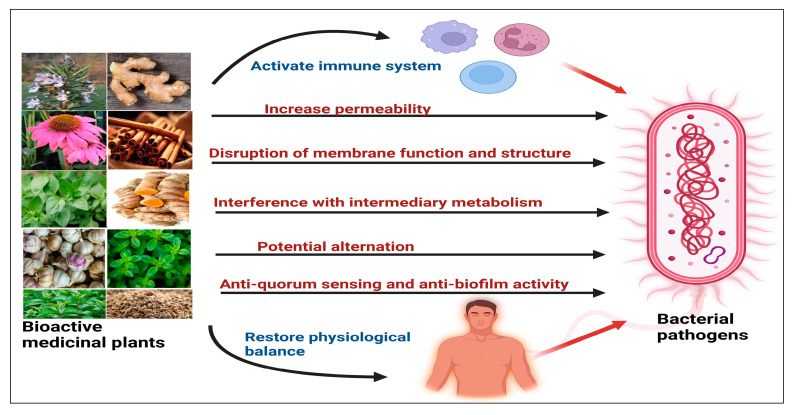
Possible antibacterial modes of action of antibacterial medicinal plants.

**Table 1 plants-12-03077-t001:** World Health Organization (WHO) global priority list of antibiotic-resistant bacteria *.

Degree of Priority	Bacterial Pathogen	Gram-Stain	Type of Resistance
Critical priority	*Acinetobacter baumannii*	G(–)	Carbapenem-resistant.
	*Pseudomonas aeruginosa*	G(–)	Carbapenem-resistant.
	Enterobacteriaceae **	G(–)	Carbapenem-resistant and third-generation-cephalosporin-resistant.
High priority	*Enterococcus faecium*	G(+)	Vancomycin-resistant.
	*Staphylococcus aureus*	G(+)	Methicillin-resistant and vancomycin-resistant.
	*Helicobacter pylori*	G(–)	Clarithromycin-resistant.
	*Campylobacter* spp.	G(–)	Fluoroquinolone-resistant.
	*Salmonella* spp.	G(–)	Fluoroquinolone-resistant.
	*Neisseria gonorrhoeae*	G(–)	Third-generation-cephalosporin-resistant and fluoroquinolone-resistant.
Medium priority	*Streptococcus pneumoniae*	G(+)	Penicillin-resistant.
	*Haemophilus influenzae*	G(–)	Ampicillin-resistant.
	*Shigella* spp.	G(–)	Fluoroquinolone-resistant.

* Tuberculosis (*Mycobacterium tuberculosis*) is not on this list since it is already a worldwide emergency in need of new treatments. ** *Escherichia coli*, *Enterobacter* spp., *Proteus* spp., *Klebsiella pneumonia*, *Providencia* spp., *Morganella* spp., and *Serratia* spp. are all in the family Enterobacteriaceae.

**Table 2 plants-12-03077-t002:** Some plants with antibacterial potential (in vitro) against pathogens considered in the priority list by WHO reported in the last four years.

Plant to Obtain the Extract/Part of the Plant	Extract or Compound Tested with Effective Action	Bioactive Compounds	Mechanism of Action	Isolates Exhibiting Superior Outcomes	MIC	Reference
*Matayba oppositifolia*/bark	Aqueous extract Hexane extract Ethyl acetate Methyl extract	Palmitic acid, friedelan-3-one, 7-dehydrodiosgenin.	-	Carbapenem-resistant *A. baumannii* Carbapenem-resistant *K. pneumoniae* Carbapenem-resistant *P. aeruginosa* Carbapenem-resistant *Enterobacter* spp.	250–1000 µg/mL	[142]
*Curcuma longa*/rhizome	Aqueous extract	Turmeric and chitosan	-	Carbapenem-resistant *P. aeruginosa*	1024 µg/mL	[143]
*Andrographis paniculate*/leaves	Ethyl acetate extract	Terpenoids and saponins	-	Carbapenem-resistant *A. baumannii*, β-Lactamase producing *E. coli*	250–500 µg/mL 25 μg/mL	[144] [145]
*Momordica Balsamina*/fruit	Methyl extract	-	-	Carbapenem-resistant *A. baumannii*	0.5 mg/mL	[146]
*Artocarpus heterophyllus*/seed	Hexane extract	-	-	Multidrug-resistant *P. aeruginosa*	125 mg/mL	[147]
*Schinus terebinthifolia*/leaves	Pentagalloyl glucose	-	-	Carbapenem-resistant *A. baumannii* Carbapenem-resistant *P. aeruginosa*	16–256 µg/mL	[148]
*Cissus incisa*/leaves	α-Amyrin-3-Oβ-D- glucopyranoside Cerebrosides mixture	-	-	Carbapenem-resistant *P. aeruginosa*	100 μg/mL	[141]
*Paeonia lactiflora*/roots	Paeoniflorin	C_23_H_28_O_11_	Breach of membrane integrity	Carbapenem-resistant *K. pneumoniae*	1200 µg/mL	[149]
*Hechtia glomerata*/leaves	Hexane Extract Aqueous extract β-sitosterol β-sitosteryl acetate daucosterol, daucosteryl acetate	-	-	Carbapenem-resistant *K. pneumoniae* Carbapenem-resistant *P. aeruginosa* Carbapenem-resistant *A. baumannii* *E. coli* ESBL	100–500 µg/mL	[150]
*Khaya senegalensis*/bark *Tamarindus indica*/bark	Aqueous extract Ethyl extract Methyl extract	-	-	Carbapenem-resistant *E. coli*	25–400 mg/mL	[151]
*Solanum chrysotrichum*/leaves	Hexane extract Dichloromethane fraction Steroidal saponins	-	-	Carbapenem-resistant *P. aeruginosa* Carbapenem-resistant *A. baumannii*	125–250 µg/mL	[152]
*Avicennia marina*/leaves	Ethanolic extract	Flavonoids, phenolics, triterpenes, and glycosides		Vancomycin-resistant *E. faecalis*	4.0 mg/mL	[153]
*Illicium verum*/seeds	Essential oils	Phenolics and flavonoids	Produce permanent damage to the cell membrane and cell contents	Methicillin-resistant *S. aureus* (MRSA)	0.25–1.0 µg/mL	[154]
*Laureliopsis philippiana*/leaves	Essential oils	Eucalyptol, linalool, isozaphrol, isohomogenol, α-terpineol, and eudesmol	-	*Helicobacter pylori* (clinical isolates)	64.0 µg/mL	[155]
*Origanum Compactum*/areal parts *Lavandula stoechas*/areal parts	Essential oils	-	Bactericidal and anti-biofilm formation	Multidrug-resistant *Campylobacter* spp.	0.063% (*v*/*v*)	[156]
*Salvia officinalis*/leaves	Ethanolic extract	-	-	Multidrug-resistant *Helicobacter pylori*	3.1–50.0 mg/mL	[157]
*Stryphnodendron adstringens*/bark	Ethanolic extract	Polyphenols and tannins	-	*N. gonorrhoeae* ATCC 49226 *K. pneumoniae* ATCC13693 MRSA (clinical isolate) *S. pneumoniae* ATCC 6303	3.125 mg/mL 12.5 mg/mL 3.125 mg/mL 0.78 mg/mL	[158]
*Cinnamomum verum*/inner bark	Essential oils	Cinnamaldehyde dimethyl acetal, cinnamaldehyde, and α-copaene	Bactericidal and inhibit bacterial DNA gyrase and topoisomerase	*S. enterica* (clinical isolate) *E. coli* ATCC 25922	0.5% *v*/*v* 0.25% *v*/*v*	[159]
*Thymus vulgaris*/areal parts	Essential oils	Phenolic monoterpenes, sesquiterpenoids (β-caryophyllene), phenylpropanoids, aliphatics, furanoids, and diterpenes.		*H. influenzae* ATCC 49247 *S. aureus* ATCC 29213	512.0 µg/mL 512.0–1024.0 µg/mL	[160]
*Litsea cubeba*/undefined	Essential oils	-	Production of reactive oxygen species and destruction of the cell membrane	*Shigella sonnei* ATCC 25931 *Shigella sonnei* CMCC 51592	4.0 μL/mL 6.0 μL/mL	[161]
*Acacia Senegal*/leaves	Hydroethanolic extract	Phenolic compounds, flavonoids, and tannins	-	Multidrug-resistant *E. coli* Multidrug-resistant *K. pneumoniae*	256.0 μg/mL >512.0 μg/mL	[162]
*Moringa oleifera*/seeds	Essential oils	Phenolic compounds and flavonoids	Bactericidal	*Helicobacter pylori* (clinical isolates)	0.5 μg/mL	[163]

## Data Availability

Not applicable.

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
