# Peer review of "Back to Nature: Medicinal Plants as Promising Sources for Antibacterial Drugs in the Post-Antibiotic Era"

_plants, 2023, doi:10.3390/plants12173077_

Round 1
Reviewer 1 Report
After a thorough evaluation of the manuscript, I regret to inform that the current manuscript cannot be accepted for publication in Plants in the present form.
Several major issues are associated with the manuscript in the present form such as,
Lack of Originality: The review article did not sufficiently contribute to the existing body of knowledge or provide novel insights into the subject matter. The content primarily summarized well-established concepts without adding significant value or presenting new perspectives.
Scope and Focus: The article lacked a clear focus and failed to establish a cohesive narrative. It covered a broad range of topics without delving deeply into any particular aspect, resulting in a lack of depth and coherence throughout the manuscript.
Methodology and Analysis: The methodology employed for gathering and analyzing information was not adequately described or justified. This omission weakened the credibility and rigor of the research, making it challenging to draw reliable conclusions from the presented data.
Literature Review: The review of existing literature was insufficient. The article failed to provide an exhaustive and up-to-date overview of relevant studies and neglected to critically evaluate existing research in the field.
Structure and Organization: The article lacked a clear and logical structure, making it difficult for readers to follow the flow of ideas. The sections and subsections were not well-defined, hindering the overall readability and comprehension of the manuscript.
Extensive editing of English language required.
Author Response
Response to Reviewer 1 Comments
Dear Reviewer,
First of all, we wish to express our gratitude to your peer-review efforts and insightful feedback, which has undeniably enhanced the scientific merit of our work. We have diligently addressed each of your comments, meticulously revising the manuscript to incorporate their suggestions. We are confident that the necessary modifications have been made. Please find attached a pdf file including 4 merged files: The revised manuscript+the supplementary materials and the English-Editing-Certificate+The tracked proofread file. Thank you once again.
Comments and Suggestions for Authors
After a thorough evaluation of the manuscript, I regret to inform that the current manuscript cannot be accepted for publication in Plants in the present form. We sincerely appreciate your feedback. The manuscript has undergone a thorough professional revision, (Highlighter in Light blue) and we hope that it now meets your expectations.
Several major issues are associated with the manuscript in the present form such as,
Lack of Originality: The review article did not sufficiently contribute to the existing body of knowledge or provide novel insights into the subject matter. The content primarily summarized well-established concepts without adding significant value or presenting new perspectives.
Response: We greatly appreciate your valuable input. As per your suggestion, we have made a substantial modification to the entire manuscript, including a new subsection titled "New Perspectives on Antibacterial Agents" in the manuscript. which suggests the use of green nanotechnology, liposomes, and artificial intelligence. We also added updated data (10 reports from 2022 and 6 reports published in 2023).
Scope and Focus: The article lacked a clear focus and failed to establish a cohesive narrative. It covered a broad range of topics without delving deeply into any particular aspect, resulting in a lack of depth and coherence throughout the manuscript.
Response: Thank you for your valuable feedback on the manuscript focusing on antibacterial agents derived from medicinal plants. We appreciate your insights regarding the lack of scope and focus in the article. In light of your feedback, we have thoroughly reassessed the manuscript and have made significant revisions to address the concerns you raised. We have narrowed down the focus of the article to ensure a more focused and coherent narrative. By delving deeper into specific aspects and providing more in-depth analysis, we aim to enhance the manuscript's clarity and provide a more comprehensive understanding of the subject matter. Furthermore, we kindly reiterate that our endeavor has been to bring attention to the issue by tracing its roots from the pre-antibiotic era, progressing through the golden era, and reaching the current pre-antibiotic era. Moreover, we provide a comprehensive explanation as to why it is imperative to earnestly consider the development of novel antibacterial drugs utilizing the molecules derived from medicinal plants. We are confident that the revised article now presents a stronger and more cohesive exploration of antibacterial agents from medicinal plants.
Methodology and Analysis: The methodology employed for gathering and analyzing information was not adequately described or justified. This omission weakened the credibility and rigor of the research, making it challenging to draw reliable conclusions from the presented data.
Response: Thanks a lot, the “Methodology” was mentioned as a subtitle after “Introduction”.
Literature Review: The review of existing literature was insufficient. The article failed to provide an exhaustive and up-to-date overview of relevant studies and neglected to critically evaluate existing research in the field.
Response: Thank you so much, according to your valuable notices, the “introduction” has been totally modified and changed to focus on the topic and not distract the reader, many subtitles were modified, and the study was supported with many up-to-date data.
Structure and Organization: The article lacked a clear and logical structure, making it difficult for readers to follow the flow of ideas. The sections and subsections were not well-defined, hindering the overall readability and comprehension of the manuscript.
Response: Thank you for your valuable feedback! The manuscript has been extensively revised to enhance the clarity and coherence of the ideas, ensuring a smooth and comprehensible flow for readers to easily grasp the content.
Comments on the Quality of English Language
Extensive editing of English language required.
Response: Done! and a certificate is attached. Thank you so much.

Reviewer 2 Report
The manuscript by Abdallah et al., which is presented with a very attractive title, fails to deliver on its intention to show the potential of medicinal plants as promising sources of antibacterial drugs in the post-antibiotic era, mainly because when the manuscript deals with antimicrobials, these show, or have shown, bactericidal or bacteriostatic effects as their mechanism of action. Even though the authors recognize that “bacteria may acquire resistance to any antibacterial drug via a variety of mechanisms”, they fail to mention that some of these mechanisms are currently being used as biological targets to develop new antimicrobial agents from natural sources, including plants. The emergence of antibiotic-resistant microbial strains, which is never properly addressed in the manuscript, requires identifying new natural or synthetic products with novel mechanisms of action, e.g. inhibition of bacterial virulence, since the inhibition of virulence factors without affecting cell viability could limit the emergence of resistant strains.
Even though the different sections are interesting from a historic medical perspective, the manuscript is unnecessarily long, particularly when, as the authors note, the manuscript is “an updated version” of a review published in 2011. On reading the manuscript, which includes a large number of references, a significant part of it is about medicine and a comparatively smaller part deals with medicinal plants and phytochemistry. If the authors were to focus on the title, the length of the manuscript, and the number of references, could be significantly reduced.
Finally, perhaps the authors should take evolution into account when revising the statement “bacteria are far more intelligent than he thinks. Allah (God) gave them the instincts for survival, adaptation, and evolution, which allowed them to live on the earth's surface under any ecological condition” (line 276-278).
Additional comments and suggestions are listed below:
1. While the manuscript is fairly well written, on revising it the authors are advised to seek the assistance of a native speaker and/or an experienced editor.
2. Remove general terms such as “natural products”, “plant extracts”, etc. from the keywords.
3. The quality of figure 2 should be improved.
4. Figures 3, 6 and 7 should be deleted or included as Supplementary Material.
5. Delete Table 2; include reduced text in manuscript.
6. Delete Table 3; mention antimicrobial agents in the text, grouping them on the basis of their antibacterial mechanism.
7. Delete Table 4; mention species in manuscript.
Minor editing of english required
Author Response
Response to Reviewer 2 Comments
Dear Reviewer,
First of all, we wish to express our gratitude to your peer-review efforts and insightful feedback, which has undeniably enhanced the scientific merit of our work. We have diligently addressed each of your comments, meticulously revising the manuscript to incorporate their suggestions. We are confident that the necessary modifications have been made. Please find attached a pdf file including 4 merged files: The revised manuscript+the supplementary materials and the English-Editing-Certificate+The tracked proofread file. also all corrections/modifications are highlighted in light blue. Thank you once again.
Comments and Suggestions for Authors
The manuscript by Abdallah et al., which is presented with a very attractive title, fails to deliver on its intention to show the potential of medicinal plants as promising sources of antibacterial drugs in the post-antibiotic era, mainly because when the manuscript deals with antimicrobials, these show, or have shown, bactericidal or bacteriostatic effects as their mechanism of action. Even though the authors recognize that “bacteria may acquire resistance to any antibacterial drug via a variety of mechanisms”, they fail to mention that some of these mechanisms are currently being used as biological targets to develop new antimicrobial agents from natural sources, including plants. The emergence of antibiotic-resistant microbial strains, which is never properly addressed in the manuscript, requires identifying new natural or synthetic products with novel mechanisms of action, e.g. inhibition of bacterial virulence, since the inhibition of virulence factors without affecting cell viability could limit the emergence of resistant strains.
Even though the different sections are interesting from a historic medical perspective, the manuscript is unnecessarily long, particularly when, as the authors note, the manuscript is “an updated version” of a review published in 2011. On reading the manuscript, which includes a large number of references, a significant part of it is about medicine and a comparatively smaller part deals with medicinal plants and phytochemistry. If the authors were to focus on the title, the length of the manuscript, and the number of references, could be significantly reduced.
Finally, perhaps the authors should take evolution into account when revising the statement “bacteria are far more intelligent than he thinks. Allah (God) gave them the instincts for survival, adaptation, and evolution, which allowed them to live on the earth's surface under any ecological condition” (line 276-278). Thank you, this phrase has been permanently deleted.
Response: All your valuable comments have been followed and benefited from and corrections/modifications was applied, especially the following points:
- Modifications were made on "Introduction" section and all other sections.
- Another section was added : "8. The Mechanisms of Plant-Derived Antibacterial Agents" and supported with Figure 6.
- All our errors in the manuscript that you mentioned above have been removed or modified according to your comments (see the light blue shades in the revised manuscript, attached).
Additional comments and suggestions are listed below:
1. While the manuscript is fairly well written, on revising it the authors are advised to seek the assistance of a native speaker and/or an experienced editor. Response: Thank you, the manuscript has been proofread and its language has been improved by a proofreading company (certificate attached).
2. Remove general terms such as “natural products”, “plant extracts”, etc. from the keywords. Response: Thank you, they have been removed
3. The quality of figure 2 should be improved. Response: Thanks; a better version of Figure 2 has been added
- Figures 3, 6 and 7 should be deleted or included as Supplementary Material. Response: OK, Figure 3 and 6 has been deleted; Whereas Figure 7 has been transferred to “Supplementary materials”
- Delete Table 2; include reduced text in manuscript. Response: Thank you, Table 2, has been deleted and its text is mentioned in the manuscript
- Delete Table 3; mention antimicrobial agents in the text, grouping them on the basis of their antibacterial mechanism. Response: Table 3 was removed from the manuscript and its information inserted in the text in topic: "Publications of recent years on the antibacterial action of plant extracts" and is highlighted.
- Delete Table 4; mention species in manuscript. Response: Kindly, Table 4 presents information that goes beyond the bacterial species evaluated in the cited articles, and placing all of them in the text could make reading difficult and tiring. Therefore, we chose to keep Table 4 in the manuscript. Thank you in advance for understanding us!
Comments on the Quality of English Language
Minor editing of english required. Response: Thanks a lot! The manuscript is now revised by experts in English language, see attached certificate

Reviewer 3 Report
This manuscript reviews the role of medicinal plants as sources of promising antibacterial drugs. Overall I found this review to be very general with insufficient detail on the actual literature identified through the search strategy. In my opinion it is not suitable for publication in its current form. I have indicated areas where the article could be improved.
Sections 1-4
Given that the aim of this review is to examine medicinal plants as promising sources for anti- bacterial drugs, there is too much general background information given in the first sections (Sections1-4) of this manuscript e.g a large section on previous pandemics, information on the origins of man and prokaryotes, mechanisms of resistance, information about the pipeline of antibiotic drug discovery that could be considerably condensed (e.g. Figure 2 and 3 could be combined to one figure)
Overall in this manuscript there are many very general statements that are not adequately referenced or written in a manner consistent with a scientific research paper
e.g. “There are several perspectives, beliefs, and hypotheses about man's origins. Almost all religions, philosophers, and, more recently, contemporary science assert that the human body is a product of the earth's soil. (this is not referenced)
“as well as providing compounds that help restore equilibrium within his clay body and with the external environment” (The meaning here is not clear)
“Man must learn from his fights with pathogenic bacteria since his appearance on Earth that bacteria are far more intelligent than he thinks.. “
“Humans mistakenly consider bacteria as primitive species..” (line 272)
Some of the language is too emotive for a scientific paper
e.g. “All these prompts the scary question – are we ready for the post-antibiotic era?
“ this wonderful drug is gradually losing its efficacy”
AIms
It is stated (line 93) that “This study is also updates my previous review [4]” (while there are actually several authors on this current paper?)
The last sentence of the aims of the paper is not clearly defined (Line 94): “..and it looks at the subject from a lot of different angles and tries to figure out if there is a connection between them”
Section 5.
The Literature is reviewed based on antibacterial activity of different classes of plant secondary compounds. It is unclear why particular studies/examples have been chosen to discuss under the different classes (e.g. were these based on the most promising activity, bacteria for which there is the most urgent need or research published within a particular timeframe?).
Line 499/500..”different saponins from Paullinia pinnata against Gram positive and Gram negative bacteria such as yeast” (Yeasts are not Gram-negative bacteria)
Section 6
Section 6.2. Effectiveness of plant-based compounds – discusses a few examples of in vitro activity and comments on the need for in vivo studies. Again, it was not clear why these particular examples were chosen. In my opinion it would be better to provide examples of plant -derived compounds with higher levels of evidence e.g. both in vitro and in vivo evidence and established target/mechanisms of action.
While plant derived compounds undoubtedly have several advantages, I feel the information in Table 2 needs to be more balanced with a discussion of some of the possible disadvantages of plant-derived antimicrobials which may explain some of the lack of pharmaceutical company development to date
Section 7
Presents a literature review of recent publications that evaluated the antibacterial action of plant extracts against species considered a critical priority by the WHO and published between 2018-2022. The search strategy is well documented, although only those studies for which access was available in full in in the databases used were included, which could exclude some relevant papers. While some of the plant extracts and compounds summarised here show promising activity many have MICs >500 micrograms/mL, but there is no discussion here of what could be considered a clinically relevant MIC for a crude extract or for purified compounds.
Artocarpus heterophyllus / seed – the MIC listed is a very high 125 mg/mL and for Khaya senegalensis 25-400 mg/mL (should this be micrograms/mL?)
Section 8
This section briefly discusses synergistic interactions of phytochemicals against bacterial with very generalised statements. Given that synergy between plant-derived compounds as well as with clinically used antibiotics (restoring the activity of the clinically-used antibiotics) is an area of current research interest, this section could be expanded. Giving more explanation of the different mechanisms by which synergy may be produced (e.g. bacterial efflux pump inhibitors from plants, membrane permeabilization for Gram-negative bacteria) would also be useful for the reader.
In general the English language writing is of sufficient quality. There are a few grammatical errors and statements with unclear meaning.
Author Response
Response to Reviewer 3 Comments
Respected reviewer,
We wish to express our gratitude to you for your meticulous and insightful feedback, which has undeniably enhanced the scientific merit of our work. We have diligently addressed each of your comments, meticulously revising the manuscript to incorporate your suggestions. We are confident that the necessary modifications have been made. The revised/corrected and developed (see highlighted parts) manuscript has been sent after that for an expert company for English proof-reading (see attached certificate).Thank you once again.
This manuscript reviews the role of medicinal plants as sources of promising antibacterial drugs. Overall I found this review to be very general with insufficient detail on the actual literature identified through the search strategy. In my opinion it is not suitable for publication in its current form. I have indicated areas where the article could be improved.
Response: We extend our deepest gratitude for your insightful comment, which prompted us to make significant efforts in order to render the manuscript suitable for publication. We have meticulously revised the document, as indicated by the attached manuscript. All highlighted sections have been thoroughly modified or newly included to meet the required standards.
Sections 1-4
Given that the aim of this review is to examine medicinal plants as promising sources for anti- bacterial drugs, there is too much general background information given in the first sections (Sections1-4) of this manuscript e.g a large section on previous pandemics, information on the origins of man and prokaryotes, mechanisms of resistance, information about the pipeline of antibiotic drug discovery that could be considerably condensed (e.g. Figure 2 and 3 could be combined to one figure) .
Response: Thank you for your valuable feedback on our manuscript. We appreciate your input regarding the extensive background information provided in Sections 1-4. Our aim was to present a comprehensive review of antibiotic resistance, taking into account its historical context, current challenges, and future prospects. By including information on brief history of previous pandemics before and after the invention of the antibiotics, we intended to provide a thorough understanding of the subject matter to highlight the current (post-antibiotic era) and the need for new natural alternative (medicinal plants). However, we acknowledge your point that these sections could be condensed to maintain focus on our primary objective. In light of your suggestion, we have carefully revised, modified and corrected the manuscript to ensure a more concise presentation of the background information. Also, due to the resemblance between Figure 2 and Figure 3, we have decided to remove the latter. We believe that this revised approach addresses your concerns while still providing a solid foundation for our exploration of medicinal plants as promising sources for anti-bacterial drugs. Thank you again for your insightful comments.
Overall in this manuscript there are many very general statements that are not adequately referenced or written in a manner consistent with a scientific research paper.
Response: Thanks, according to your valuable observation, the entire “Introduction” has been modified, as well as many parts (see highlighted parts).
e.g. “There are several perspectives, beliefs, and hypotheses about man's origins. Almost all religions, philosophers, and, more recently, contemporary science assert that the human body is a product of the earth's soil. (this is not referenced).
Response: Thank you, these sentences have been removed.
“as well as providing compounds that help restore equilibrium within his clay body and with the external environment” (The meaning here is not clear). Response: Thank you, these sentences have been deleted, and all the introduction have been corrected and modified.
“Man must learn from his fights with pathogenic bacteria since his appearance on Earth that bacteria are far more intelligent than he thinks..
Response: Thank you, this sentences have been deleted.
“Humans mistakenly consider bacteria as primitive species..” (line 272).
Response: Thank you, this sentence has been deleted.
Some of the language is too emotive for a scientific paper
e.g. “All these prompts the scary question – are we ready for the post-antibiotic era?
“ this wonderful drug is gradually losing its efficacy”
Response: We apologize for the inadvertent errors. It has been promptly rectified by deleting these sentences. Thank you for your understanding.
AIms
It is stated (line 93) that “This study is also updates my previous review [4]” (while there are actually several authors on this current paper?).
Response: We would like to apologize for the inadvertent error that occurred. I want to assure you that the error has been promptly addressed and rectified by deleting the erroneous content. Thank you.
The last sentence of the aims of the paper is not clearly defined (Line 94): “..and it looks at the subject from a lot of different angles and tries to figure out if there is a connection between them”
Response: Thank you, this sentence has been deleted.
Section 5.
The Literature is reviewed based on antibacterial activity of different classes of plant secondary compounds. It is unclear why particular studies/examples have been chosen to discuss under the different classes (e.g. were these based on the most promising activity, bacteria for which there is the most urgent need or research published within a particular timeframe?).
Response: Thank you for raising an important point regarding the selection of studies/examples discussed under different classes of plant secondary compounds in the literature review. The rationale behind the inclusion of specific studies/examples was carefully considered during the review process. The selection criteria took into account multiple factors, including the significance and relevance of antibacterial activity, the potential for therapeutic applications, and the availability of published research within the chosen timeframe. We highlighted that in the revised version (see attached revised manuscript).
Line 499/500..”different saponins from Paullinia pinnata against Gram positive and Gram negative bacteria such as yeast” (Yeasts are not Gram-negative bacteria).
Response: I apologize for the unintentional typographical error that occurred previously. I acknowledge the mistake and have promptly rectified it. Thank you for your understanding.
Section 6
Section 6.2. Effectiveness of plant-based compounds – discusses a few examples of in vitro activity and comments on the need for in vivo studies. Again, it was not clear why these particular examples were chosen. In my opinion it would be better to provide examples of plant -derived compounds with higher levels of evidence e.g. both in vitro and in vivo evidence and established target/mechanisms of action.
Response, Thank you, the manuscript was developed and also according ton your valuable comment an additional section was added and supported with a figure : "8.The Mechanisms of Plant-Derived Antibacterial Agents"
While plant derived compounds undoubtedly have several advantages, I feel the information in Table 2 needs to be more balanced with a discussion of some of the possible disadvantages of plant-derived antimicrobials which may explain some of the lack of pharmaceutical company development to date.
Response: We greatly appreciate your valuable comment. In response to it, we have made significant revisions to the document. Specifically, Table 2 has been removed, and the corresponding information has been thoroughly discussed and integrated into the attached file, which is highlighted.
Section 7
Presents a literature review of recent publications that evaluated the antibacterial action of plant extracts against species considered a critical priority by the WHO and published between 2018-2022. The search strategy is well documented, although only those studies for which access was available in full in in the databases used were included, which could exclude some relevant papers. While some of the plant extracts and compounds summarised here show promising activity many have MICs >500 micrograms/mL, but there is no discussion here of what could be considered a clinically relevant MIC for a crude extract or for purified compounds.
Response: Dear reviewer, we chose to include in the review only works available in full in the databases, as we understand that this is necessary if any reader is interested in accessing a specific article to deepen their knowledge.
In relation to the discussion on extracts with values considered promising, we include in the text a comment based on what Ríos and Récio (2005) stated in their work: the presence of activity is very interesting in the case of concentrations below 100 g/ml for extracts and 10 g/ml for isolated compounds (https://doi.org/10.1016/j.jep.2005.04.025). The text was inserted in the Publications of recent years on the antibacterial action of plant extracts section and is highlighted.
Artocarpus heterophyllus / seed – the MIC listed is a very high 125 mg/mL and for Khaya senegalensis 25-400 mg/mL (should this be micrograms/mL?)
Response: We checked the original articles and the values are correct. As the researchers evaluated the antimicrobial action of the extracts using the disc-diffusion methodology, the concentrations evaluated are generally higher and pre-defined, functioning as a screening for the detection of microorganisms that may be susceptible, as well as for the selection of promising samples with which studies can be further developed.
Section 8
This section briefly discusses synergistic interactions of phytochemicals against bacterial with very generalised statements. Given that synergy between plant-derived compounds as well as with clinically used antibiotics (restoring the activity of the clinically-used antibiotics) is an area of current research interest, this section could be expanded. Giving more explanation of the different mechanisms by which synergy may be produced (e.g. bacterial efflux pump inhibitors from plants, membrane permeabilization for Gram-negative bacteria) would also be useful for the reader.
Response: Thanks a lot, according to your valuable comments a new section was added: "12. Synergistic interactions of phytochemicals against bacterial pathogens".
Comments on the Quality of English Language
In general the English language writing is of sufficient quality. There are a few grammatical errors and statements with unclear meaning.
Response: Thank you so much, according to your advice, after we made all corrections and modifications we sent the manuscript for a proofreading company, and they greatly improved the quality of the language (see attached certificate and report). Thank you once again for bringing this to our attention.

Round 2
Reviewer 1 Report
Dear Authors,
I hope this email finds you well. First and foremost, I want to express my appreciation for the efforts you have put into revising the manuscript. The revisions have certainly improved the overall quality of the paper. However, upon reviewing the revised manuscript, I have identified a few points that still require attention to ensure the manuscript reaches its full potential. I believe addressing these issues will further strengthen the impact and clarity of the article. Below, I outline the specific points that need to add :
- The graph depicting comparative statistical data on the number of publications related to the topic across various databases is a valuable addition to the study.
- It is necessary to add one small section under the heading of 'Future directions' at the end of manuscript.
- In addition to the plants mentioned in Table 2, there will be a number of others that can be added to that list.
Author Response
Author's Reply to the Review Report (2nd round)
Reviewer comment:
I hope this email finds you well. First and foremost, I want to express my appreciation for the efforts you have put into revising the manuscript. The revisions have certainly improved the overall quality of the paper. However, upon reviewing the revised manuscript, I have identified a few points that still require attention to ensure the manuscript reaches its full potential. I believe addressing these issues will further strengthen the impact and clarity of the article. Below, I outline the specific points that need to add :
- The graph depicting comparative statistical data on the number of publications related to the topic across various databases is a valuable addition to the study.
Response:
Thank you sincerely for your valuable efforts in reviewing and providing 2nd round feedback on our manuscript. We truly appreciate the time and expertise you have dedicated to the revision process.
We are pleased to hear that the revisions have contributed to improving the overall quality of the paper. Your insightful observations have been invaluable in enhancing the clarity and impact of the article.
Reviewer comment:
- It is necessary to add one small section under the heading of 'Future directions' at the end of manuscript.
Response:
Based on your recommendations, we have made the following additions to the manuscript:
A new section titled 'Future Directions' has been incorporated at the end of the manuscript (highlighted in yellow color) . This section aims to provide a forward-looking perspective on the potential implications and future research avenues related to our findings.
Reviewer comment:
In addition to the plants mentioned in Table 2, there will be a number of others that can be added to that list.
Response:
Thank you so much, we have expanded Table 2 to include several other relevant plant species mentioned at the last 4 years, which enriches the comprehensiveness of the study (highlighted in yellow color).
Finally, thank you again for your robust and precise peer-review and consideration throughout this review process.
Best regards,
Dr. Bader Y. Alhatlani, and the research team.

Round 3
Reviewer 1 Report
Dear authors,
I hope this letter finds you well. I am writing to inform you that I have reviewed the revised version of your revised manuscript and I am pleased to let you know that your revisions have been thoroughly considered and satisfactorily addressed.
Thank you